# Recent Advances in Covalent Drug Discovery

**DOI:** 10.3390/ph16050663

**Published:** 2023-04-28

**Authors:** Daniel Schaefer, Xinlai Cheng

**Affiliations:** 1Buchmann Institute for Molecular Life Sciences, Chemical Biology, Goethe University Frankfurt am Main, Max-von-Laue-Strasse 15. R. 3.652, 60438 Frankfurt am Main, Germany; schaefer@pharmchem.uni-frankfurt.de; 2Pharmaceutical Chemistry, Goethe University Frankfurt am Main, 60438 Frankfurt am Main, Germany; 3Frankfurt Cancer Institute, 60596 Frankfurt am Main, Germany

**Keywords:** covalent inhibitors, PROTACs, drug discovery, drug design, COVID-19

## Abstract

In spite of the increasing number of biologics license applications, the development of covalent inhibitors is still a growing field within drug discovery. The successful approval of some covalent protein kinase inhibitors, such as ibrutinib (BTK covalent inhibitor) and dacomitinib (EGFR covalent inhibitor), and the very recent discovery of covalent inhibitors for viral proteases, such as boceprevir, narlaprevir, and nirmatrelvir, represent a new milestone in covalent drug development. Generally, the formation of covalent bonds that target proteins can offer drugs diverse advantages in terms of target selectivity, drug resistance, and administration concentration. The most important factor for covalent inhibitors is the electrophile (warhead), which dictates selectivity, reactivity, and the type of protein binding (i.e., reversible or irreversible) and can be modified/optimized through rational designs. Furthermore, covalent inhibitors are becoming more and more common in proteolysis, targeting chimeras (PROTACs) for degrading proteins, including those that are currently considered to be ‘undruggable’. The aim of this review is to highlight the current state of covalent inhibitor development, including a short historical overview and some examples of applications of PROTAC technologies and treatment of the SARS-CoV-2 virus.

## 1. Introduction

Medical research has progressed exponentially over the last century [1]. In a lot of cases, diseases that were considered to be death sentences 100 years ago can now be cured using drugs [2]. For instance, the discovery of antibiotics has drastically increased human life expectancy and reduced the progress and severity of symptoms [1,2]. Drug discovery is centered around the development of active substances [3]. Depending on the desired result, different active substances have been designed to have distinct pharmacodynamic properties (e.g., pain relief or blood pressure reduction) [3]. Many small molecule drugs can inhibit and, therefore, prevent the biological activity of a protein of interest (POI), while a few drugs can stimulate POI activity. Regardless of the functions of small molecule drugs, the effects generally depend on the interaction between the active substance (e.g., an inhibitor, effector, or activator) and the POI (e.g., an enzyme, protein, ion channel, or receptor) [3]. These interactions can be divided into two general categories: non-covalent interactions and covalent interactions (Figure 1) [4].

Due to negative experiences with covalent-reactive compounds (especially with highly reactive drug metabolites, which can trigger immunogenicity and idiosyncratic drug reactions), covalent inhibitors did not enjoy widespread popularity until 1990 [5,6]. At that time, most research groups and companies developed active compounds with non-covalent binding properties [6]. Figure 2 displays a timeline of the development or commercialization of covalent binding agents over the years [6,7,8,9,10,11].

More recent studies have shown that chemical optimization can enhance the activity and target specificity of covalent inhibitors in clinical use, which has greatly encouraged scientists to develop more covalent inhibitors [6]. Figure 3 illustrates the number of publications in the SciFinder portal containing the terms ‘covalent drug’ and ‘inhibitor covalent’ over time.

Although covalent binders can be toxic due to the undesired modification of off-target proteins or haptenization [8,12], compounds that rely on covalent mechanisms are among the major classes of small molecules, representing about 30% of all active substances on the market. Table 1 shows the covalent inhibitors that have been approved by the FDA since 2010.

Aspirin is an essential medicine that is recommended by the WHO for anti-inflammatory, fever reduction, and pain relief functions [10,12]. Aspirin can irreversibly acetylate cyclooxygenase (COX-1 and COX-2) enzymes, which are the key catalysts in response to the formation of pro-inflammatory prostaglandins [10]. Upon transferring its acetyl group to the hydroxy group of the side chain of Ser_530_, aspirin blocks the binding of arachidonic acid (a substrate to cyclooxygenase), thereby inhibiting its activity [13]. Moreover, due to the irreversible binding, oxygenase remains inhibited until the cell is degraded [13]. The mechanism of action of aspirin is illustrated in Figure 4.

Saxagliptin [14] is an active substance that is used to treat diabetes mellitus type 2. In contrast to aspirin, which is an irreversible covalent inhibitor, saxagliptin reversibly inhibits the enzyme dipeptidyl peptidase 4 (DPP-4) via the amino acid residue Ser_630_ and is a reversible covalent inhibitor [14]. DPP-4 is a serine exopeptidase that contains a catalytic triad comprising Ser_630_, His_740_, and Asp_708_ in its binding pocket, which degrades the hormone glucagon-like peptide 1 (GLP-1) and, in turn, prevents the release of insulin [14]. Higher insulin concentrations reduce the glucagon concentration and, thus, blood glucose levels [14]. Because saxagliptin is a reversible covalent inhibitor, the duration of inhibition depends on the reverse reaction or hydrolysis of the covalent complex [14]. The mechanisms of inhibition and the release of DPP-4 are shown in Figure 5.

Whether covalent inhibition is reversible or irreversible depends on the binding energy. Generally, irreversible bonds have higher binding energy than reversible bonds [15]. Although the energy required to break a covalent bond is relatively high compared to that required to break a non-covalent bond, covalent bonds can be broken by various inter- and intramolecular chemical reactions [15]. For example, water is one of the most important reagents for such cleavage, despite its moderate nucleophilicity, due to its very high concentration [15]. Other factors include the alternation of the environmental pH value (e.g., in the stomach) [16,17] and the presence of specific enzymes that can break corresponding covalent bonds (e.g., lipase for esters and amidase for amides) [18,19,20]. For example, an ester bond can be easily hydrolyzed into alcohol and an acid [21]. In addition to substrate-dependent fission, intramolecular fission can also break covalent bonds, for instance, to form energetically lower ground states [22].

## 2. Advantages and Disadvantages of Covalent Inhibitors

The ability of covalent inhibitors to form chemical bonds with target proteins can have several advantages [23]. For instance, in many cases, the enzymatic activity of a protein is related to a non-covalent or only transiently covalent interaction with its substrate [23]. Thus, the displacement of irreversible covalent inhibitors using a natural substrate is nearly impossible [23]. In general, the dose of a drug is positively correlated to its toxicity [24]. In comparison to non-covalent inhibition, the covalent bond formation can enable full target occupancy even at relatively low concentrations [24]. In addition, covalently binding drugs are usually less susceptible to drug resistance that is caused by mutations in chemotherapy, as long as the covalent binding modes remain unaffected by the mutations [25]. However, changes that affect the formation of covalent binding often lead to drug resistance, such as mutations of the nucleophile, blockages of binding sites, or reductions in nucleophilic characteristics [6,24]. The advantages and disadvantages of covalent and non-covalent inhibitors are shown in Table 2.

For instance, the epidermal growth factor receptor (EGFR) is a surface receptor that belongs to the tyrosine kinase family [26]. Upon binding to EGF, EGFR transduces external signals to cells for proliferation. Gefitinib [26,27] is an adenosine triphosphate (ATP)-competitive protein kinase inhibitor that has a significant impact on EGFR-related signaling pathways by blocking the adenosine triphosphate (ATP)-binding sites of enzymes, resulting in the inhibition of the enzymes. However, drug resistance has often been reported due to the occurrence of various mutations during long-term treatments [28]. The most common mutation is the replacement of threonine at position 790 with methionine (T790M), which alters the binding pocket and prevents the binding [28]. T790 is known as the gatekeeper residue because the amino acid residue is critical for access to and the size of the binding pocket. The exchange of the polar amino acid threonine for the bulky nonpolar amino acid methionine leads to increased resistance to first- and second-generation EGFR inhibitors. After treatment, the receptors remain active, and tumor cells continue to proliferate [28]. In contrast, afatinib is a covalent EFGR inhibitor that can irreversibly bind to mutated and WT EGFR; however, it can lead to dose-dependent toxicity and has a stronger affinity for the wild-type EGFR [29,30]. Osimertinib is a third-generation EGFR inhibitor that shows improved selectivity and less toxicity than afatinib and has a stronger affinity to mutant EGFR than wild-type EGFR. Therefore, osimertinib can be used to circumvent the dose-limiting toxicity of second-generation inhibitors [30]. Clinical studies have reported less drug resistance to afatinib and Osimertinib [31] by tumors with EGFR^T790M^ [28,30]. The structures of gefitinib, afatinib, and osimertinib are shown in Figure 6.

However, as mentioned above, certain mutations that prevent the formation of covalent bonds, such as EGFR^C797S^, confer resistance to Osimertinib [32]. This highlights that in contrast to non-covalent drugs, clinical applications of covalent drugs still need to overcome several drawbacks [6,33]. Rapid, irreversible inhibition can be advantageous for covalent inhibitors; however, this feature can also lead to undesirable long-term effects (e.g., toxicity) when proteins are inhibited over a long period of time and are not metabolized due to the long turnover of the proteins. Nevertheless, in an impressive discussion, Juswinder Singh was able to illustrate that covalent protein kinase inhibitors do not appear to exhibit higher toxicity than non-covalent inhibitors [30]. However, it can be assumed that as long as proteins are in the system, undesired interactions can occur, as shown by the following example [33,34,35]. Clopidogrel is a prodrug for thrombosis prevention, which can inhibit the adenosine diphosphate receptor P2Y12 [34,35]. P2Y12 is a member of the inhibitory G-protein-coupled purine receptor family and promotes platelet aggregation [35,36]. In the human body, clopidogrel is metabolized through oxidation and subsequent hydrolysis [35]. During this conversion, various diastereomers are formed, among which the only active metabolite is that with (S)/(R) configuration (Figure 7) [37]. This active metabolite irreversibly inhibits P2Y12 through a reaction between the thiol group of the active metabolite (Figure 7; highlighted in blue) and the Cys_97_ side chain within the first extracellular loop of P2Y12, which forms a disulfide bridge. As a result, platelet aggregation is prevented [36].

Because this action slows down blood clotting, long-term irreversible inhibition can lead to prolonged bleeding times, resulting in certain critical consequences, especially in the case of accidents or emergency operations after taking clopidogrel [36]. Additionally, the inhibition of platelet aggregation is sometimes also related to unusual bleeding from vessels in the eyes or lungs [36].

Another factor that negatively impacts the development of covalent agents is the rapid compensation to inhibition by newly synthesized target proteins because of the homeostasis of the body upon inhibition [24]. This is particularly problematic for proteins with very high protein turnover or in cases where protein turnover is increased by disease, treatment, or other circumstances, as shown in the following examples.

In a review by Shringarpure et al., they described that oxidative stress could increase the intracellular degradation of short-lived and long-lived proteins and that progressive oxidation further increases the degradation of proteins via proteasomes [38]. In another case, Davies et al. described that Crohn’s disease leads to abnormal protein turnover [39]. It has also been shown that children with active disease have increased protein turnover [39]. With conventional treatments, protein breakdown and synthesis are reduced, resulting in no changes in net protein balance in remission [39]. In the case of rapid protein turnover, re-administration is required to reach the critical concentration for the inhibition of the target protein. However, excessive drug intake may induce severe side effects (e.g., toxicity) [24].

## 3. Mechanisms of Action and Chemical Designs

The entire process involving the interaction between a target and a covalent inhibitor up to the formation of a covalent bond takes place in two steps [6]. The first step is the reversible association between the inhibitor and the target protein [6]. In the second step, a reaction takes place that forms a covalent bond [6]. This is exemplified by telaprevir, which reversibly inhibits the viral NS3.4A protease of the hepatitis C virus (HCV; Figure 8) [40].

Often, covalent inhibitors carry electrophilic groups, which react with nucleophilic residue on the target enzymes [6,41]. The warheads can be epoxides, aziridines, esters, ketones, nitriles, or another similar group [41]. For example, penicillin is a covalent inhibitor with beta-lactam as the warhead, which reacts with the active serine residue in the D-alanine transpeptidase [11]. Transpeptidases are essential for cross-linking in the biosynthesis of bacterial cell walls [11]. Irreversible bonds inhibit transpeptidase, resulting in the lysis of bacterial cells due to their instability [11]. The mechanism of action of this irreversible inhibition is shown in Figure 9.

To date, many new functional groups have been found that can form covalent bonds with sulfur-containing functional groups, as shown in Figure 10 [42]. The advantage of these groups is that they can directly react with the cysteine in target proteins at its active site without a prior metabolic activation [42]. Figure 10 displays various warheads that are involved in the formation of irreversible and reversible bonds [42]. In most cases, inhibitors occupy the binding pockets, which prevents substrates from forming bonds (i.e., competitive inhibition) [42]. Nevertheless, occupation in the active sites of target proteins is not always necessary [42]. For instance, a few inhibitors can bind to the remote sides of target enzymes, resulting in the alternation of the binding pocket [42]. These inhibitors are called allosteric inhibitors [42]. In rare cases, uncompetitive inhibition can occur, in which inhibitors bind exclusively to enzyme–substrate complexes. This results in the formation of enzyme–substrate–inhibitor complexes, which ensures that the enzymes do not convert the substrates; therefore, no products are formed [43]. Irreversible inhibitors are divided into two types: affinity label inhibitors and mechanism-based inhibitors (suicide inhibitors) [44]. Affinity label inhibitors resemble enzyme substrates and enter the active sites of enzymes, where irreversible covalent bonds are formed, and the active sites are modified without enzymatic conversion [44]. Suicide inhibitors bind to active sites in the same way as substrates, triggering the enzymatic properties of the enzymes [44]. During the enzymatic process, intermediaries are formed that cannot be further converted or split off. As a result, no further substrates can be converted. Aspirin and penicillin are examples of suicide inhibitors [44].

The formation of a bond can have different mechanisms. Many electrophilic warheads can react with a nucleophile via Michael addition, for example, [45]. The attacking nucleophile (e.g., carbanion, amine or thiol) serves as a Michael donor and the electrophile (e.g., α,β-unsaturated carbonyl compound) as a Michael acceptor [45]. A more detailed description of the different types of covalent reactions in drug development and the associated groups has been presented in great detail by Gehringer et al. [45] and Shindo et al. [46] and is not exclusively discussed here.

The design of chemical inhibitors generally includes three essential steps [47]. The first step comprises a structural analysis of the target protein, including amino acid sequence analysis and a three-dimensional enzymatic structure analysis [47]. One of the most important aspects of this step is the identification of potentially attackable nucleophilic residues (e.g., cysteine and lysine), which are preferentially located in the binding pockets of receptors [48]. Furthermore, serine and threonine can act as potentially nucleophilic side chains, for example, when they are present as catalytic residues in proteases. Furthermore, tyrosine could be an interesting target for covalent inhibitors since the amino acid in a neutral state has a lower intrinsic nucleophilicity than cysteine or unprotonated lysine [45]. In addition, the hydroxy group is slightly more acidic than the protonated amine of lysine side chains. The deprotonation of the hydroxy group leads to the production of phenoxy anions, which have highly nucleophilic properties. Phenoxy anions are hard nucleophiles that react preferentially with hard Lewis acids; thus, they can be taken into consideration in the design of covalent inhibitors [45]. However, compared to cysteine and lysine, only a few warheads prefer tyrosine as the primary amino acid residue (e.g., sulfonyl fluorides and sulfonyl fluoride analogs), which limits the choice of warheads but can probably increase selectivity. The strength of the nucleophilicity plays a decisive role. Cysteine has the highest intrinsic nucleophilicity (pKa ≈ 8.5) among the proteinogenic amino acids; therefore, many warheads target cysteine. Lysine (pKa ≈ 10.5) and tyrosine (pKa ≈ 10) are not usually involved in catalytic triads but can be targeted when the reactive form is favored by the local environment. In contrast, the nucleophilic properties of serine (pKa > 13) and threonine (pKa > 13) is very low but can still be targeted, for example, when they are part of a catalytic triad [47,48]. When proteins have more than one reactive nucleophile (e.g., cysteine and lysine), inhibitors should ideally have a stronger affinity for the nucleophile of interest than other internal (reactive) nucleophiles. Important co-factors for bond formation are the environment and the accessibility of the nucleophile [47,48]. The second step involves the identification of known inhibitors with good potency, the binding properties and mechanisms of which are also ideally known [48]. High-throughput screening (HTS) is often used to identify potential inhibitors by testing a large number of molecules against target proteins [48,49]. However, in the field of covalent inhibitors, this procedure is not widely used, and very few covalent inhibitors have been identified using this method. A more widely used method in the field of covalent inhibitors is the fragment-based drug discovery (FBDD) [50]. Interest in this method has continued to grow over recent years as it has proven to be a highly successful method, most recently with the FDA-approved compounds sotorasib and Osimertinib [50]. Later on, the connection between FBDD and the discovery of sotorasib will be explained. While HTS usually screens large libraries of large drug-like molecules, FBDD screens smaller and less complex molecules. Studies have shown that despite their low affinity for target proteins, the hits have better ‘atom-efficient’ binding interactions than hits in HTS [50]. Fragment libraries cover a much larger chemical space than HTS libraries, although the number of compounds in fragment libraries is smaller. In addition, small fragments have much fewer opportunities to interact with target proteins compared to large molecules, thus, avoiding suboptimal interactions and enabling the identification of ‘qualitatively’ better ligands or binding partners [50]. Furthermore, it is easier to find additional binding sites, such as allosteric centers. In general, hits from fragment libraries have low dissociation constants (range: μM-mM), which distinguishes them from hits from HTS (range: nM-low μM) [50]. The hits obtained from fragment libraries can be used as lead structures either directly or with minor modifications, whereas in the case of HTS hits, the development of lead structures can be much more complex [50]. Another important factor in this process is the pan-assay interference compounds (PAINs) [49]. Frequently, these chemicals can unspecifically interact with diverse proteins and produce false positive results in high-throughput screening [51]. When a potential inhibitor is already known, and the above properties are present, docking studies are often conducted as the third step [51,52]. A major problem in docking covalent inhibitors is that, in most cases, the reactivity of warheads is not considered in the docking algorithm. However, docking can help to determine the correct position and orientation of warheads by testing different linkers, positions, and warheads (Figure 10) before chemical synthesis takes place [48]. However, results have sometimes been contradictory to real experimental outcomes [48]. In covalent reactions, the free activation energy (i.e., the barrier that must be overcome to start the reaction) determines whether inhibition is reversible or irreversible [53]. When the free activation energy is low, then the covalent bond is reversible, whereas the covalent bond is irreversible when the free activation energy is high [53,54]. Moreover, introducing steric hindrance in active sites can prevent covalent bonds from breaking and increase the residence time of ligands [54].

Activity-based protein profiling (ABPP) is a standard approach for studying interactions between inhibitors and their expected and unexpected binding partners, where target proteins are located in their physiological environments [55,56]. To perform ABPP, cells or lysates are treated with an activity-based probe and the proteins that are covalently bound to the drug can be identified [55,56]. The activity-based probe consists of a warhead, linker, and reporter tag [55,56]. Using this method, it is possible to investigate off-target effects by labelling the inhibitor with a tag, such as fluorophore or biotin [56]. When binding occurs between the labeled inhibitor and proteins beyond the primary target, this process can be investigated using further analytical procedures, depending on the tag used [55,56]. Another application of ABPP is possible through the use of an additional reactive probe, whereby inhibitors need not necessarily be labeled [56]. ABPP probes consist of two elements: a reactive group and a tag [55]. Additionally, a linker can also be incorporated between the reactive group and the tag to increase the distance between the reporter and the reactive group to avoid steric hindrance [55]. Generally, the reactive groups consist of electrophilic functional groups that form covalent bonds with the nucleophilic residues at the active sites of the enzymes [55,56]. When the active site of an enzyme is blocked by the presence of an inhibitor, the tagged molecule cannot bind, and there is no change in the signal [55,56]. Based on data obtained from ABPP, the binding properties of inhibitors can be optimized [54]. Moreover, optimization can alter selectivity, activity, half-life and other molecular properties [54]. A brief illustration for the design of covalent inhibitor is shown in Figure 11.

A more detailed description of the rational design of targeted covalent inhibitors has been presented by Lonsdale et al. [57] and is not exclusively discussed here. Various covalent inhibitors are described below.

### 3.1. Alzheimer’s Disease: Acetylcholinesterase

Acetylcholinesterase (AchE) belongs to the cholinesterase family and hydrolyzes acetylcholine to acetic acid and choline [58]. Its active site consists of a catalytic triad comprising serine, histidine, and glutamic acid, which is crucial for hydrolysis [58]. Acetylcholine (Ach) acts as a neurotransmitter for signal transmissions from neurons to end organs in the central and peripheral nervous systems [59]. In the peripheral nervous system, Ach is also responsible for the transmission of excitation from nerves to muscles [59]. Alzheimer’s disease is most likely caused by the protein deposition of extracellular amyloid plaque and intracellular neurofibrillary tangles (NFTs) in the brain [58,59]. Among other things, this results in faulty or absent signal transmission, which impacts mental and physical health [59]. One symptom of Alzheimer’s disease is the death of neurons that are responsible for producing neurotransmitters (including acetylcholine) [58]. This process decreases the concentration of neurotransmitters [58,59]. The breakdown of existing Ach can be drastically slowed by acetylcholinesterase inhibitors, which can attenuate the deterioration of a patient’s mental and physical state [59]. In contrast to tacrine [60] and donepezil [61], which are two well-known reversible non-covalent acetylcholinesterase inhibitors, rivastigmine [62] is a covalent inhibitor for acetylcholinesterase, which binds pseudo-irreversibly to Ser_203_ (esteratic site). Mechanistically, Ser_203_ first attacks the carbamate group of rivastigmine and then a phenol derivative is cleaved to form a carbamate group with the serine of the esterase (Figure 12) [62]. The carbamate group is slowly hydrolyzed by water and the function of the esterase is restored (Figure 12). The average duration of inhibition is approximately 10 h [63]. Another covalent inhibitor is metrifonate [64]. Metrifonate is an organophosphate pro-drug that non-enzymatically converts to dichlorvos (DDVP; O,O-dimethyl-O-(2,2,-dichlorovinyl) and acts as a pseudo-irreversible inhibitor, similarly to rivastigmine. The structures and mechanisms of action of these two pseudo-irreversible covalent inhibitors (rivastigmine and metrifonate (DDVP)) are shown in Figure 12.

As described earlier, the inhibition of AchE slows down the progression of symptoms in Alzheimer’s disease [58,59,63]. However, this inhibition does not prevent the death of neurons nor the deterioration of the patient’s mental and physical condition [58,59,65,66].

### 3.2. X-Linked Agammaglobulinemia [67], B-Cell Leukemia [68], and B-Cell Lymphoma [69]: Bruton’s Tyrosine Kinase (BTK) [70,71]

Bruton’s tyrosine kinase (BTK) is a non-receptor tyrosine kinase that belongs to the TEC family [71], is expressed by B-cells and plays a crucial role in B-cell maturation [71]. When errors occur during maturation or cell division, the body tries to break down the faulty fragments or cells [71]. When defective cells cannot be recognized, B-cell leukemia develops in the bone marrow and blood or B cell lymphoma develops in the lymph nodes, depending on the site of the defective cell division [68,69,70]. In addition, BTK malfunction can cause Bruton’s disease [72]. The first manufactured BTK inhibitor was ibrutinib [73], which was approved by the FDA in 2013. Ibrutinib provided the first option for the chemotherapy-free treatment of B-cell malignancies [74]. Unfortunately, off-target side effects and emerging resistance to ibrutinib treatment have been observed. As a result, the second-generation inhibitors acalabrutinib [75] (2017) and zanubrutinib [76] (2019) were approved. These inhibitors irreversibly bind BTK covalently to the Cys_481_ of the ATP-binding pocket [74]. Acalabrutinib has been shown to have the lowest off-target rate and the highest selectivity, followed by zanubrutinib and ibrutinib. These inhibitors differ in terms of pharmacodynamics and pharmacokinetics, which influence clinical practice in terms of their dosing and efficacy [74]. Dosing is relevant for BTK occupancy. Ibrutinib has a higher half-life than acalabrutinib and is generally only administered once daily. BTK occupancy studies have shown that acalabrutinib achieves greater BTK occupancy with twice daily administration due to its reduced half-life [74]. For zanubrutinib, twice daily administration also shows greater BTK occupancy. A reduction in the potential risk of off-target effects and the rapid inhibition of target proteins can be obtained by balancing rapid absorption with rapid elimination [74]. The shorter half-life and selective properties of acalabrutinib can lead to the complete and continuous inhibition of BTK without increasing the risk of off-target effects by inhibiting other kinases. Furthermore, the complete occupation of BTK has been shown to reduce drug resistance [74]. Compared to ibrutinib and zanubrutinib, which carry αβ-unsaturated carbonyl moiety, acalabrutinib has a propiolamide electrophilic warhead [75]. The structures of the three inhibitors are shown in Figure 13.

### 3.3. Pancreatic Cancer [77], Colorectal Cancer [78], and Lung Cancer [79,80]: KRAS G12C Mutation [81]

KRAS is a monomeric G-protein that is part of the central intermediate element in signal transduction during the growth and differentiation phases of cells [82]. The activity of KRAS is tightly controlled by GTPase-activating proteins (GAP). KRAS is only functionally active in the GTP-bound state [81,82]. The hydrolysis of GTP to GDP results in the inactivation of KRAS [81]. The point mutation of the KRAS gene impairs GTP hydrolysis, resulting in the stabilization of the activated GTP-RAS form [82]. In this case, KRAS remains permanently active [81], leading to the accumulation and uncontrolled growth and differentiation of cells, which can lead to tumorigenesis [82]. Among various mutations, the G12C mutation is one of the most important variations in lung, colon and pancreatic cancers [81,83,84]. Interestingly, a non-native reactive cysteine has been found in this mutation, which has inspired researchers to develop covalent inhibitors [85]. In 2013, Ostrem et al. [85] published a large number of different inhibitors that can form covalent bonds with the KRAS G12C mutation. Initially, the authors used tethering compounds on a disulfide basis, which were determined via a library screen using protein mass spectroscopy. Furthermore, using co-crystal structures, they showed that the tethering compounds form disulfide bridges with Cys_12_. In this case, the inhibitors are not located in nucleotide pockets but rather in allosteric sites next to them, which largely consist of switch-II and, therefore, are called switch-II pockets. Instead of further investigating these disulfide-based compounds, the authors then turned to carbon-based electrophiles, more specifically acrylamides and vinylsulfonamides (Figure 14). These warheads remain chemoselective but form irreversible bonds with Cys_12_ [85]. A comparison of the co-crystal structures between a tethering compound and compound 8 (Figure 14) showed that both covalent inhibitors react with Cys_12_ but the switch-I and switch-II regions shift to different extents [85]. While for the tethering compound, there is only a slight change in the switch-II and switch-I regions, compound 8 caused the significant displacement of switch-II, resulting in the disordering of switch-I [85]. The shift in these regions leads to a lack of density in the metal ion (Mg^2+^), the coordination of which is crucial for nucleotide binding [85]. Therefore, the mutation of the magnesium-coordinating residues Ser_17_ and Asp_57_ has significant effects on the preferences for GDP or GTP, with GDP being preferred by the mutation [85]. The authors performed further studies and were able to show that the mutation of the metal bond and the associated change in nucleotide affinity due to binding to the switch-II binding pocket did not result in the exclusion of Mg^2+^ [85]. Therefore, the inhibition leads to the preferential binding to GDP and the inactivation of G12C [85]. One year later, another approach was reported by Hunter et al. [86], in which the inhibitors were structurally derived from GDP and could irreversibly inhibit G12C. The SML-8-73-1 inhibitor has an electrophilic chloroacetamide warhead and has been shown to promote the inactive form of G12C through binding [86]. This covalent binding is detectable via mass spectrometry [86].

However, further experiments showed that the compound is cellularly impenetrable due to two negative charges on the phosphate, which lead to a modified structure (SML-10-70-1) [86]. The warheads of both inhibitors are α-chloroacetamide. The use of α-chloroacetamides or α-haloacetamides should be treated with particular caution because, depending on the type of leaving group and the steric environment (especially the substituent at the α-position), the warheads can become very reactive [45]. The reaction of α-haloacetamides with, for example, cysteine follows an S_N_2 mechanism. The reactivity of α-iodoacetamide is extremely high and drops when iodine is replaced by bromine. In a publication by Gehringer et al., the reactivity of α-haloacetamides was investigated using a GSH assay to measure the half-lives at a pH of 7.4 and 37 °C [45]. The half-life of α-bromoacetamide was found to be 0.08 h [45]. When the bromine is replaced by chlorine, the half-life increases to 3.2 h. The reactivity of α-haloacetamides can be modified by other factors, such as substituents at the α-position and nucleophiles. For example, when a methyl group is present at the α-position, this increases the half-life in the GSH assay to >60 h, indicating lower reactivity [45]. Furthermore, the reactivity of α-haloacetamides can also be reduced by adding steric bulk near the reaction site. It has also been shown that the reactivity of α-chloroacetamide and acrylamide warheads is in the same range [45]. These structures are displayed in Figure 15.

Based on the work of Ostrem et al. and Hunter et al., a number of covalent inhibitors were reported in the following years with improved selectivity, activity and cell penetration. Currently, there are several acrylamide-containing KRAS G12C inhibitors in clinical trials [87], which have shown high selectivity due to enhanced reactivity with mutated cysteine, which is located in the switch-II pocket of KRAS G12C. The switch-II pocket is centered between the central β-sheet and switch-II and shows significant conformational changes after ligands bind to the pocket [87]. Following the discovery of the switch-II binding pocket, potent inhibitors were found through the use of FBDD that bind to Cys_12_ and served as the fundament for sotorasib. Remarkably, it took only eight years from the first publication by Ostrem et al., demonstrating covalent binding to G12C, to FDA approval of sotorasib. One reason for the short time between discovery and FDA approval was the use of FBDD. On 28 May 2021, and 12 December 2022, sotorasib and adagrasib were approved by the FDA, respectively [88,89]. Both compounds are administered orally and act as potent irreversible small molecule inhibitors against the KRAS G12C mutation by covalently binding Cys_12_ without affecting the wild-type KRAS protein. The recommended dosage is 960 mg daily for sotorasib and 600 mg twice daily for adagrasib [88,89]. The median time required to reach C_max_, which is the maximum plasma concentration after administration, is 1 h for sotorasib and 6 h for adagrasib [88,89]. The half-life of sotorasib is 5 h, while that of adagrasib is 23 h [88,89]. Both inhibitors have side effects, such as diarrhea, nausea, and fatigue [88,89]. The results of an adagrasib monotherapy study analyzed the DNA sequencing of tumor biopsy samples and revealed that 17 of the 38 patients developed resistance to adagrasib via various mechanisms [88,89]. The reasons for the resistance included secondary mutations or amplifications in KRAS and alternative oncogenic alterations that activated the RTK RAS pathway but did not directly alter KRAS. The emergence of secondary mutations results in differential resistance to the inhibitors. The secondary mutations Y96D and Y96S result in resistance to both inhibitors. Furthermore, G13D, R68M, A59S, and A59T result in high resistance to sotorasib, and Q99L confers resistance to adagrasib; however, interestingly, the mutations do not confer resistance to the other inhibitor [88,89]. Sotorasib is metabolized by CYP3As and adagrasib is specifically metabolized by CYP3A4 via oxidative metabolism [88,89]. It is important to mention that the metabolism of adagrasib via CYP3A4 only occurs following single administration and adagrasib inhibits its own metabolism following multiple administration. In this case, CYP2C8, CYP1A2, CYP2B6, CYP2C9, and CYP2D6 lead to the metabolism of adagrasib [88,89]. Figure 16 shows examples of new G12C inhibitors (sotorasib [88], adagrasib [89,90], GDC-6036 [91] and JNJ-74699157 [92]).

## 4. Covalent PROTACs

Because of the lack of well-defined binding pockets, fewer than 10% of disease-causing proteins can currently be targeted by chemical inhibitors [93]. Hence, new methods are continuously being sought to modulate the activity of ‘undruggable’ proteins [93], including proteolysis targeting chimeras (PROTACs) and molecular glue degraders [93,94,95]. PROTACs and molecular glue degraders have similar functional backgrounds [94]. Both can degrade ‘undruggable’ proteins by inducing protein-E3 ligase interactions [94].

The synthesis of covalent molecular glues is an emerging but so far poorly explored field [96]. An example of a covalent molecular glue was recently presented by Daniel Nomura’s research group, whose work included the conversion of protein targeting ligands into molecular glue degraders [97]. In that study, various components were modified into the solvent-free end of the CDK4/6 inhibitor ribociclib [98] and a compound containing a trifluoromethylphenyl cinnamamide, which emerged as a possible candidate. The compound, designated EST1027, results in a >50% reduction in CDK4/6 levels in C33A cervical cancer cells via proteasome-mediated degradation [97]. Furthermore, the use of bortezomib [99], a proteasome inhibitor, demonstrated that degradation occurs via proteasome. The importance of trifluoromethylphenyl cinnamamide in the degradation of CDK4/6 was then investigated by substituting trifluoromethylphenyl [97]. The results showed that no degradation was induced. Mechanism studies have revealed that EST1027 reacts with the Cys_32_ residue of RNF126 (a RING family ubiquitin ligase) via a 1,4-addition. RNF126 ligase is an important component in cellular protein quality control and ubiquitination, as well as the associated degradation of misplaced proteins in cytosol [97]. In further studies, it has been observed that a significant decrease in RNF126 concentration occurs following treatment with EST1027. Based on this research, trifluoromethylphenyl cinnamamide was tested to establish whether it could act as a general building block for the use of covalent molecular glues by modifying the CDK4/6 inhibitor Palbociclib [100] (FDA approved) [97]. Unfortunately, the results showed no binding to RNF126 nor the degradation of CDK4 in vivo [97]. Another interesting example of a covalent molecular glue was described by Słabicki et al. [101]. Their correlative analysis identified CR8, an (*R*)-roscovitine analog CDK12 inhibitor that possesses a pyridyl residue at the solvent-exposing end and forms a complex with CDK12-cyclin k and the CUL4 adaptor protein DDB1. The complex formation leads to the ubiquitination and subsequent degradation of cyclin k. It has been shown that this degradation occurs solely due to the additional pyridyl moiety, because (*R*)-roscovitine does not lead to the degradation of cyclin k [101].

In general, PROTACs are heterobifunctional molecules that consist of chemical binding sections and E3 ligase recruiters, which are connected with linkers [93,94,95,102]. In many cases, optimal linkers can improve the selectivity, flexibility, solubility, cell permeability, pharmacological (pharmacokinetic/pharmacodynamic) profile, and degradation efficiency of PROTACs [102,103]. However, the effects of linkers are associated with a number of factors and are currently almost unpredictable [102]. Commonly used linkers that have been shown to have good biological functionality are polyethylene glycol (PEG) units [103]. Furthermore, an inflexible linker was compared to a PEG unit and showed that a reduction in the degree of freedom can lead to better pharmacological properties (e.g., improved degradation), in which case linear aliphatic linkers or PEG unit can be replaced by piperine- or piperazine-based linkers [103]. Compared to molecular glue degraders (MW typically < 500 g/mol), PROTACs can vary drastically in size, ranging from 500 g/mol to >1000 g/mol [94,102]. Additionally, molecular glues and PROTACs differ in terms of their affinity to target proteins and E3 ligases [94,95,102]. Molecular glue degraders often display a very weak affinity to target proteins and E3 ligases individually [102]. The affinity is dramatically increased as soon as a ternary complex is formed [104]. On the other hand, each binding section within a PROTAC has a high affinity and selectivity to its corresponding binding proteins [94,102]. The Binding affinities might actually be reduced upon the formation of a ternary complex [95].

One of the advantages of PROTACs over conventional inhibitors is that target protein are degraded upon binding and complete resynthesis is necessary [105]. In addition, the degradation of the target by a PROTAC does not result in side effects, which can occur with conventional inhibitors due to the presence of the inhibited protein. For small molecule inhibitors, a high dose is required to induce their effect at a high occupancy, which is often associated with off-target effects [106]. In contrast, PROTAC-mediated degradation is an event-driven process. Some of the most important advantages of reversible covalent PROTACs are their catalytic properties [105,106]; however, certain prerequisites are necessary for covalent PROTACs to exhibit these catalytic properties. As mentioned above, PROTACs must bind reversibly to target proteins, but it does not matter whether the PROTACs bind reversibly non-covalently, reversibly covalently, or irreversibly covalently to E3 ligases [106,107]. The most important factor that determines the catalytic properties of PROTACs is the relationship between the dissociation rate and the degradation rate of the proteasome [106,107]. Therefore, reversible PROTACs can only possess catalytic properties when the dissociation rate is faster than the degradation rate of the target protein. Consequently, PROTACs are not subject to 1:1 stoichiometry [106,107], which drastically reduces the required dose and the associated off-target effects [107]. In addition, reversible covalent protein adducts have been found to be less cytotoxic than stable conjugates and transient adducts do not trigger the activation of cell damage signaling pathways, while stable adducts can [105,106,107]. On the other hand, their disadvantages include the limited availability of ligands that bind E3 ligases and the lack of chemical binding sections that target transcription factors or other proteins, with unknown chemical binders [106]. In comparison to non-covalent inhibitors, covalent and non-covalent PROTACs can bind to target proteins without defined binding pockets and PROTACs generally require a much lower affinity than inhibitors [107]. Analogous to covalent inhibitors, the binding sections of covalent PRTOACS can be changed and optimized [105,106]. Moreover, covalent PROTACs possess high ligand efficiency and are small in size, as well as having high activity against drug resistance [105,106,107]. Thus, covalent PROTACs exhibit favorable properties in terms of absorption, distribution, and toxicity [105]. Irreversible covalent binding to E3 ligases allows multiple targets to be recruited for ubiquitination and degradation without having to repeat the kinetic formation process of the E3-PROTAC complex each time [105,106,107]. One disadvantage of reversible covalent binding is complicated kinetic characterization [107]. Theoretically, almost any E3 ligase can be used for degradation [108]. However, currently, very few E3 ligase recruiters have been reported [108]. Among them, cereblon (CRBN) and Hippel–Lindau tumor suppressor (VHL) are two preferred ligases that are used in the design of PROTACs [109]. CRBN serves as a substrate binding unit for the CUL4 ubiquitin ligase complex [108]. Thalidomide and its derivates, such as lenalidomide and pomalidomide, are a class of CRBN binder [110]. The second E3 ligase is VHL, which is an essential component in the complex of VHL, elongin B, elongin C, and cullin-2 [109]. The VHL protein plays a crucial role as a substrate binding domain and can be adapted for the production of PROTACs via the VHL ligand VH032 [109,111].

Interesting examples of PROTACs that covalently bind to the E3 ligase and lead to degradation of the target protein are presented in the publication by Tao et al. [112]. The authors screened azetidine acrylamides in cysteine-directed MS-ABPP. Among the hits found, some compounds showed binding to Cys_1113_ of the cullin 4-RING E3 ligase (CRL4) substrate receptor DCAF1. The compound with the strongest reactivity with DCAF1-Cys_1113_ is present as an enantiomer and the configuration was shown to be crucial for binding to DCAF1-Cys_1113_ [112]. Subsequently, the active conformation served as an E3 ligase binding domain for the development of covalent PROTACs. The active conformation was linked via a PEG-based linker to the small molecule SLF, which is a high-affinity ligand for FKBP12 [112]. The results showed that the obtained electrophilic PROTAC promotes the formation of a ternary complex between endogenous DCAF1 and FKBP12 and induces ubiquitination of FKBP12. Nevertheless, no degradation of the protein was observed when endogenous DCAF1 was used, possibly due to counteracting cellular deubiquitinases [112]. Furthermore, when recombinant expressed DCAF1 was used, it was shown that only about 20% of Cys_1113_ was occupied, reinforcing the assumption. In addition, electrophilic PROTACs were linked to the corresponding DCAF1 binding moiety via a PEG linker containing JQ1 (BRD4 ligand). FKBP12 and BRD4 PROTACs were found to degrade the target protein when DCAF1-WT-expressing HEK293T cells were used [112].

In 2019, Zeng et al. [113] presented cereblon-based covalent degraders for the KRAS protein that used various linkers. They demonstrated that PROTACs display strong cell penetration and the ability to bind to the KRAS-G12C protein in vivo, although the degradation of KRAS is not sufficient [113]. The authors aim to further investigate whether better degradation of the target protein can be achieved using reversible covalent PROTACs or non-covalent PROTACs. The structure of the proposed PROTAC is shown in Figure 17.

In the same year, Xue et al. achieved the synthesis of ibrutinib-based covalent BTK-PROTACs [114]. In their work, they used the covalent inhibitor ibrutinib (Figure 13). For the E3 ligase binding moiety, they used both pomalidomide and the VHL ligand VH032. The authors attempted to connect the BTK inhibitor and the E3 ligase binder using various linkers. The authors replaced the acrylamide warheads of two covalent PROTACs with non-covalently binding propanamide units to make a direct comparison. In addition to the ibrutinib-based PROTAC, a PLS-123 (BTK inhibitor)-based PROTAC was also tested. A competition assay using fluorescent samples demonstrated that the covalent PROTACs showed reduced binding between BTK and the fluorescent samples. The same results could not be obtained with the non-covalent PROTACs. Furthermore, the inhibition of BTK autophosphorylation was detected in cellular activity assays, even after the covalent PROTAC including medium was replaced with fresh medium, due to the irreversible properties of the PROTACs. In addition, direct comparisons were conducted between the structurally analogous PROTACs in terms of their degradation activity. The first comparison showed that the covalent ibrutinib-based PROTAC had a significantly decreased BTK level versus the non-covalent PROTAC. In the case of the PLS-123-based PROTAC, comparative results were obtained between the covalent and non-covalent PROTACs. As expected, the same results were observed when measuring IC_50_ [114]. The PROTAC illustrated in Figure 18 showed the best degradation at a concentration of 3 μM, which caused the level of BTK to fall below 20% [114].

CRISPR/Cas9 technology is a powerful tool for genome and epigenome editing within basic research and clinical applications [115]. In principal, Cas9 is an endonuclease and is able to cut DNA at desired places under the guidance of gRNA [115]. Recent results from initial phase I clinical trials have demonstrated that CRISPR/Cas9-modified T cells are tolerated well by patients [115]. However, mutations have been found, which are probably caused by DNA being cut at unwanted places (one of the major safety concerns for the clinical applications of CRISPR technologies) [115]. Compelling evidence has shown a positive correlations between off-target effects and overactive Cas9 proteins, suggesting that the precise control of Cas9 protein activity can reduce off-target effects [115]. In 2016, Zhang et al. reported that the (Phe)-Cys(C)-Pro(P)-Phe(F), π-clamp) moiety is recognized by perfluoro aromatic molecules in vitro for antibody modification [116]. Gama et al. adapted this technology into Cas9 protein and demonstrated that Cas9 tagging FCPF could form a covalent bond with perfluoro aromatic compound (Figure 19) in cells [117]. They managed to degrade the Cas9^FCPF^ protein within 8 h using a lenalidomide-conjugated perfluoro aromatic compound, called FCPF-PROTAC [117]. Interestingly, FCPF-PROTAC can also sufficiently degrade other Cas proteins tagged with FCPF, including dCas9^FCPF^, dCas12^FCPF^, and dCas13^FCPF^, suggesting that this approach could be generally applied to regulate the stability of exogenously expressed proteins [117].

## 5. Covalent Inhibitors That Target the Main SARS-CoV-2 Protease M^pro^

Since December 2019, the global COVID-19 (coronavirus disease 2019; SARS-CoV-2) pandemic has claimed the lives of several million people [118]. Currently, several vaccines exist for this virus, and many more are currently in clinical trials [119,120]. These vaccines are intended to reduce the viral rate of infection and protect against severe symptoms [118]. However, a major problem with this process is the constant mutation of the virus, which reduces the protective effects of the vaccines [119]. Currently, most published inhibitors are covalent and only a few are non-covalent [121]. The warheads of covalent inhibitors largely include aldehydes, alpha-ketoamides, nitriles, acrylamides, Michael acceptors, and chloroacetamides, which form covalent bonds with active Cys_145_ residues [121]. The fact that formation at Cys_145_ is possible was discovered early on, thus, forming a good basic structure for the development of new active ingredients. Furthermore, as already mentioned, covalent drugs are more resistant to mutations, as long as the reactive nucleophiles remain intact and accessible [121]. Although great progress has been made, no active substance has yet shown satisfactory efficacy against all of the different variants of the virus in clinical trials [119].

Covalent agents are presented in three recent publications [122,123,124]. In a publication by Kokic et al., they showed that treatment with remdesivir, nirmatrelvir, or a combination of both can reduce the symptoms of COVID-19 [122]. Remdesivir [125] is an RNA-dependent RNA polymerase inhibitor, while nirmatrelvir [126] is an inhibitor against the main SARS-CoV-2 protease M^pro^, which is essential for replication via the cleavage of polyproteins [127,128,129,130,131]. Alongside ritonavir [127], which is a CYP3A4 inhibitor, nirmatrelvir was the first agent to be approved against COVID-19. The SARS-CoV-2 genome encodes two polyproteins (‘pp1a’ and ‘pp1ab’), which are cleaved by M^pro^, producing cleavage products that are essential for replication [127]. M^pro^ is a cysteine protease that cleaves glutamine (Gln) residues and consists of two coupled units that fold independently: (1) the catalytic site with the chymotrypsin-like domain ‘I + II’ with the catalytic triad of C_145_ + H_41_; (2) a cluster of helices domain ‘III’ [127]. The domains form a dimer via the interaction between the N-terminus of domain ‘I + II’ and the C-terminus of domain ‘III’ [127]. This dimer is reversible and is more stable upon binding with a substrate [127]. Because the cysteine protease that cleaves glutamine is unknown, M^pro^ has emerged as an attractive target [127,132,133,134]. M^pro^ has a glutamine preference at the P1 site, making glutamine an obvious choice as the basic unit for developing an inhibitor to target this protease [127]. The problem is that the amine of the glutamine residue could react with the trifluoroacetyl group of nirmatrelvir, resulting in a cyclized product [127]. Cyclization leads to the altered structures of inhibitors, which can result in the complete loss of their inhibitory properties. To circumvent this problem, pyrrolidone is chosen as the side chain in inhibitors, such as nirmatrelvir, to mimic glutamine [127]. In nirmatrelvir, nitrile acts as a warhead because it is a known pharmacological warhead that targets serine and cysteine proteases and is relatively inert compared to other electrophiles, which means that only very strong nucleophiles can be considered for reaction [127]. In addition, the position of nitrile also plays an important role in selectivity [127]. The introduction of the nitrile group significantly increases oral bioavailability [127]. Preclinical studies have shown few off-target effects in vitro as there are no interactions with a broad range of G-protein coupled receptors, transporters, kinases, or other enzymes [127]. Furthermore, nirmatrelvir has been tested against several HIV proteases and the IC_50_ values obtained are above 100 μM [127]. In addition, no mutagenic or clastogenic effects have been detected [127]. Reversible covalent bonds are formed between nirmatrelvir and the cysteine from the catalytic triad at position 145 of the protease [127]. In the first step, a proton transfer from Cys_145_ to His_41_ occurs and in the following step, a negatively charged sulfide attacks the nitrile group of the inhibitor [127]. In the third step, a proton transfer occurs from His_41_ to a water molecule and then another transfer occurs from this to the nitrogen atom of the nitrile group [127]. As a result, a thioimidate product is formed [127]. The crystal structure shows that, as well as the covalent bonds, there are further interactions between the inhibitor and the protease [127]. The imine nitrogen from the thioimidate forms hydrogen bonds with Gly_143_ and the Cys_145_ residue [127]. In addition to the interactions mentioned above, there are also hydrophobic interactions between the cyclopropyl structure and various hydrophobic residues of the protease [127]. The trifluoroacetyl group folds in the S4 sub-pocket, allowing the trifluoromethyl group to form hydrogen bonds with Gln_192_ [127]. The nitrogen from the same group also forms hydrogen bonds with Glu_166_. Currently, there is no known resistance to nirmatrelvir [127]. The structure of nirmatrelvir is shown in Figure 20.

In June 2022, Pillaiyar et al. [123] studied a series of covalent inhibitors (Figure 21) that can form reversible or irreversible covalent bonds with M^Pro^, thereby blocking the protease from binding to the replicase polyprotein for transcription and stagnating replication [123]. The thiol residue of Cys_145_ attacks inhibitors at the carbon atom of the thioester group and as a result, the sulfur atom of the thioester group, together with the residue (the residue in Figure 21: 3c = 1,3,4-thiadiazole; 3w + 3a = pyrimidine), acts as a leaving group. A new thioester linkage forms between the inhibitor and Cys_145_ [123]. Furthermore, no enzyme release occurs with the 3c and 3w inhibitors, so it is assumed that the inhibitors inhibit irreversibly. Inhibitor 3af shows a time-dependent inhibition in that the inhibitor is released slowly and the enzyme activity recovers [123]. Of the approximately 40 inhibitors tested in the study, some bind Cys_145_ irreversibly and some bind it reversibly. In addition, the IC_50_ value or % inhibition (at 10 μM) was measured. The IC_50_ values of the three inhibitors are shown in Figure 21 [123]. Excellent antiviral activity has been demonstrated in cell-based experiments [123]. The crystal structures discovered by Pillaiyar et al. shows covalent binding to the protease at Cys_145_ [123].

In the same year, Kneller et al. [124] published three different covalent hybrid inhibitors consisting of a splicing component from a hepatitis protease inhibitor and a known SARS-CoV-1 protease inhibitor. In that study, boceprevir [135] and narlaprevir [136] were used as hepatitis protease inhibitors. The SARS-CoV-1 protease was the M^Pro^ protease, and the inhibitors GC-376 and nirmatrelvir were used as scaffolds [124].

The inhibitors shown in Figure 22 are reversible and demonstrate antiviral activity [124]. To target cysteine at position 145, BBH-1 utilizes a ketone group, while a nitrile domain is applied in the cases of BBH-2 and NBH-2 [124]. The inhibitors tested by Kneller et al. and Pillaiyar et al. demonstrate potent inhibitory effects against COVID-19 by covalently binding to M^Pro^ [136], even in mutated variants as long as Cys_145_ is present [124].

## 6. Outlook

In recent decades, numerous new covalent inhibitors have been identified that have significantly improved selectivity and activity [42]. In comparison to non-covalent inhibitors, covalent inhibitors have several advantages in chemotherapy, including decreased drug resistance and relatively low doses [23,24]. Especially in the field of cancer research, the number of covalent inhibitors is increasing [42,93,137]. Therefore, covalent inhibitors could continue to be an important area of research in the field of drug discovery [4,6,25]. Moreover, selectivity and toxicity, during long-term treatment, are still two major concerns in the clinical application of covalent inhibitors [33], which need to be further investigated. PROTACs and related technologies are also rapidly growing areas in drug discovery. The development of new covalent warheads could support the design of covalent PROTACs to address ‘undruggable’ proteins [102,114]. This trend could also spread to various other research areas in the future.

## Figures and Tables

**Figure 1 pharmaceuticals-16-00663-f001:**
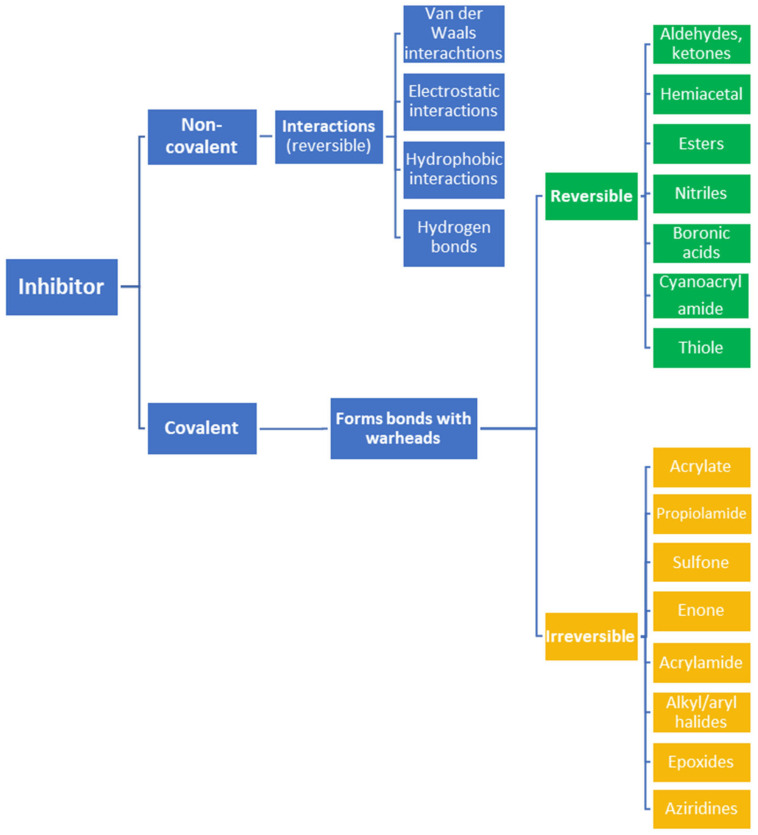
Different types of inhibitors and their potential interactions and bindings, as well as the corresponding functional groups for the formation of reversible and irreversible covalent bonds.

**Figure 2 pharmaceuticals-16-00663-f002:**
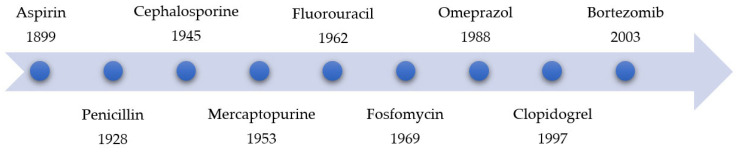
The timeline of the development of various covalent inhibitors with the associated years of discovery.

**Figure 3 pharmaceuticals-16-00663-f003:**
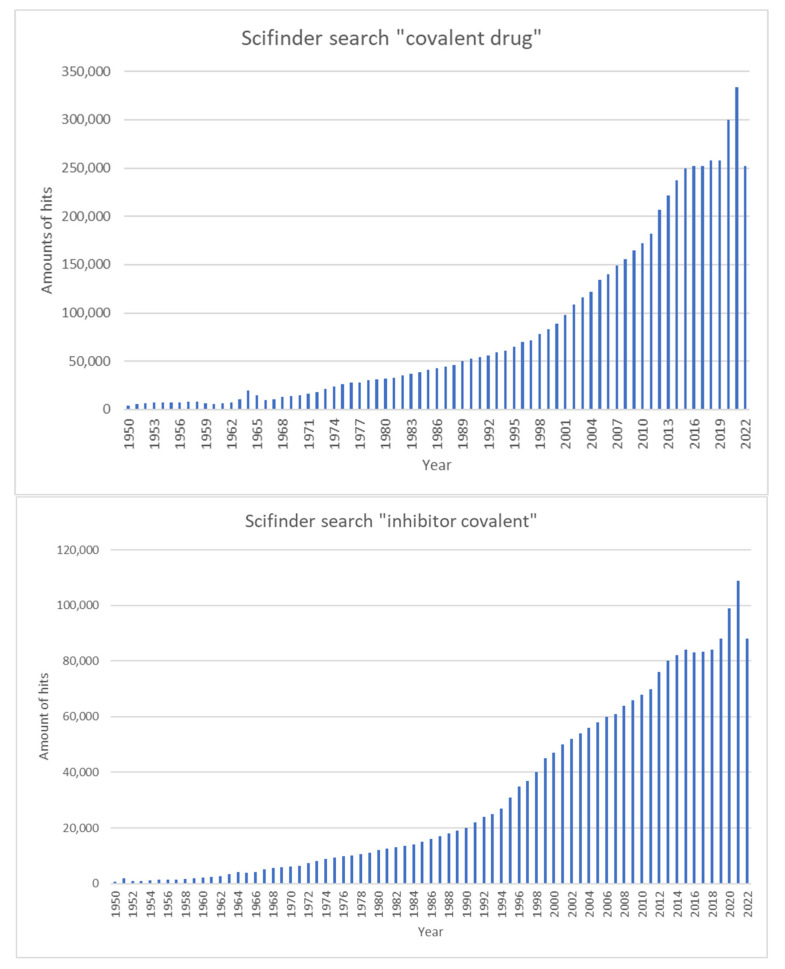
The significant increase in publications associated with covalent inhibitor development in the years from 1950 to 2022. The graphs show the number of publications with the keywords of ‘covalent drug’ (**top**) or ‘inhibitor covalent’ (**bottom**) in the SciFinder portal over the last 72 years.

**Figure 4 pharmaceuticals-16-00663-f004:**
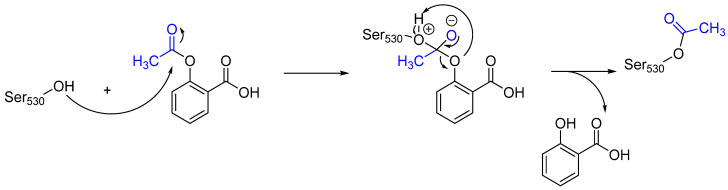
The mechanism of action of aspirin via the irreversible inhibition of COX-1 and COX-2. The reactive acetyl group of aspirin reacts with the hydroxy group of the Ser_530_ side chain, which is shown in blue.

**Figure 5 pharmaceuticals-16-00663-f005:**
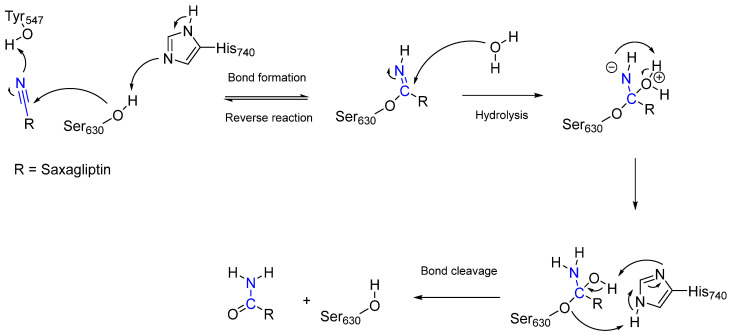
A graphical representation of the covalent inactivation of DPP-4 via the reaction of the nitrile group (highlighted in blue) with the catalytic Ser_630_ using the example of saxagliptin. The reverse reaction and the cleavage of the covalent bond between saxagliptin and DPP-4 complex by water are also shown.

**Figure 6 pharmaceuticals-16-00663-f006:**
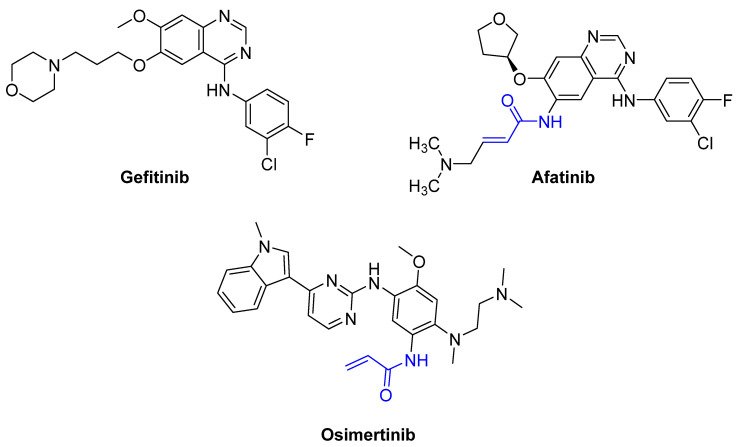
The structures of the selected first, second and third-generation EGFR inhibitors. Gefitinib (**left**) interacts non-covalently with receptors, while afatinib (**right**) and osimertinib (**bottom**) form covalent bonds with the thiol group of the cysteine side chain of the receptors. The warheads of afatinib and osimertinib (marked in blue) are responsible for the binding to the receptors.

**Figure 7 pharmaceuticals-16-00663-f007:**
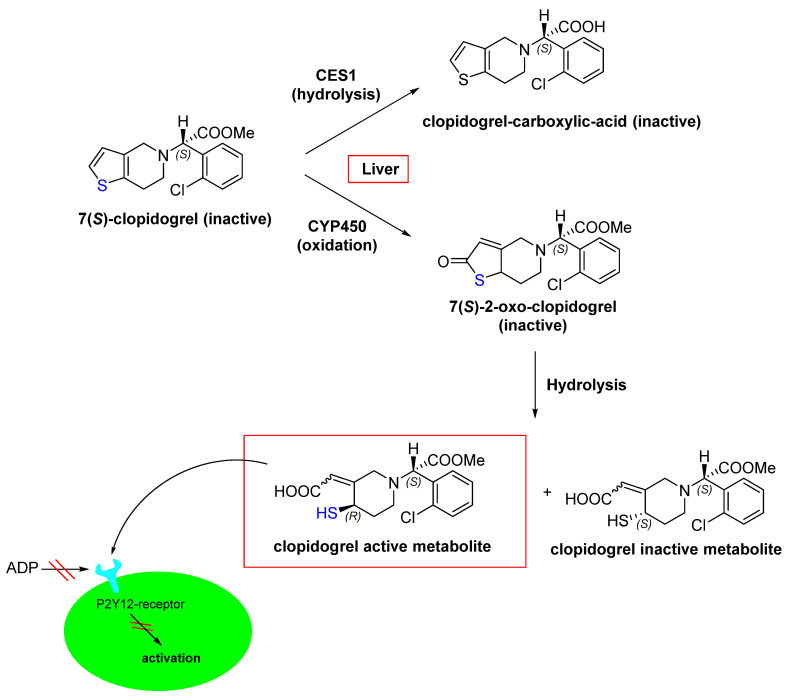
The metabolic pathway of clopidogrel (in)activation, considering the configuration required for the covalent inhibition of the P2Y12-receptor. CES1 (carboxylesterase 1) catalyzes the hydrolysis of 7(S)-clopidogrel into clopidogrel carboxylic acid, which is inactive. Cytochromes P450 (CYP450) are oxidoreductases and enable the oxidation of 7(S)-clopidogrel in the first step. CYP1A2, CYP2B6, CYP2C9, CYP2C19, and CYP3A4/5 are implicated as cytochrome P450 enzymes involved in the metabolism of clopidogrel. The warhead is highlighted in blue.

**Figure 8 pharmaceuticals-16-00663-f008:**
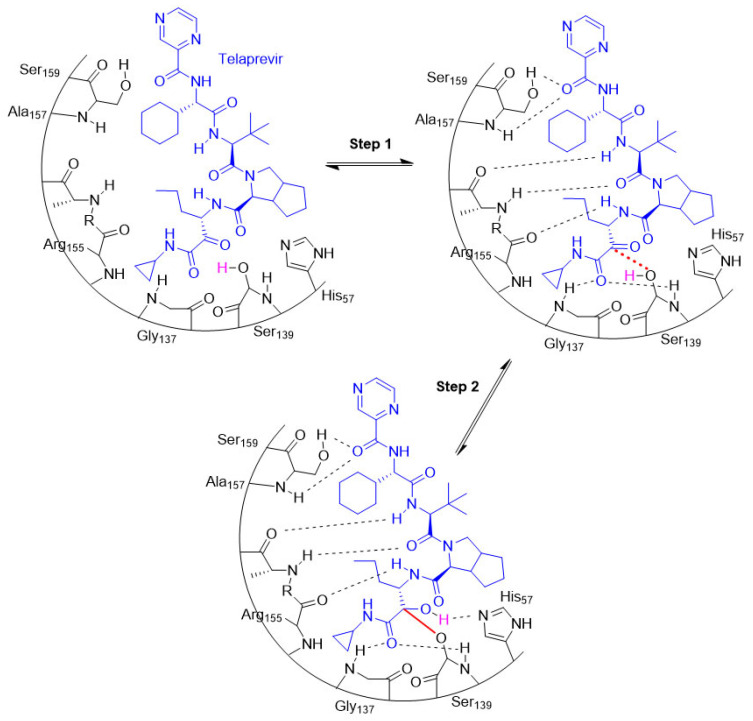
An illustration of the entire two-step process (i.e., association and bond formation) using telaprevir, the example HCV protease inhibitor. Telaprevir inhibits the viral NS3.4A protease of the hepatitis C virus. The structure of telaprevir is shown in blue.

**Figure 9 pharmaceuticals-16-00663-f009:**
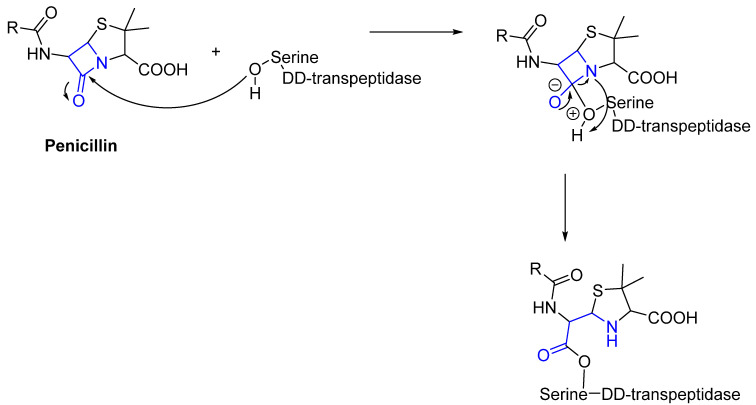
The mechanism of action of the irreversible inhibition of DD-transpeptidase by penicillin. The warhead of penicillin (β-lactam; highlighted in blue) reacts with the serine side chain of DD-transpeptidase.

**Figure 10 pharmaceuticals-16-00663-f010:**
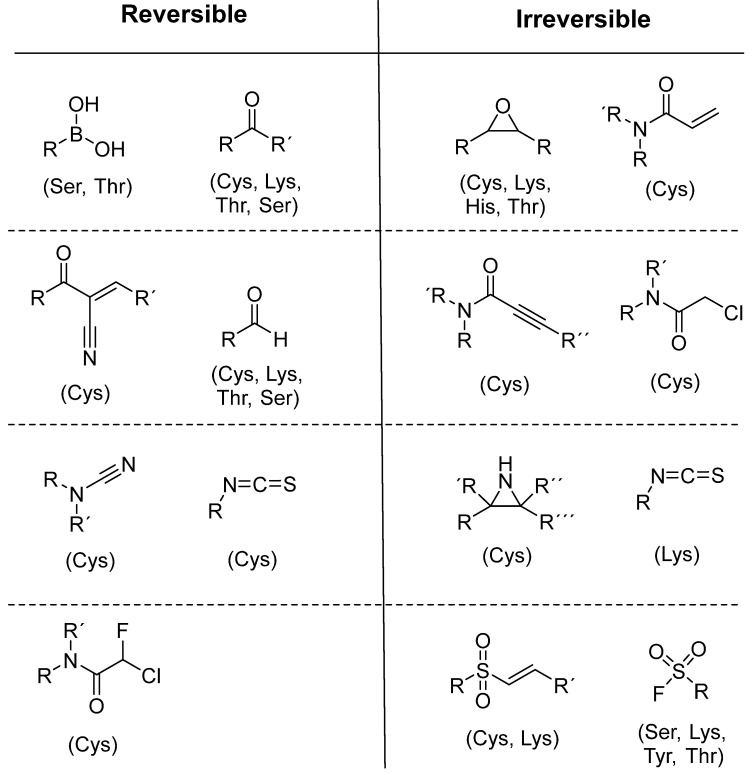
Warheads form irreversible and reversible bonds. The primary targeting amino acid residues are shown in brackets below the respective structures [45,46].

**Figure 11 pharmaceuticals-16-00663-f011:**
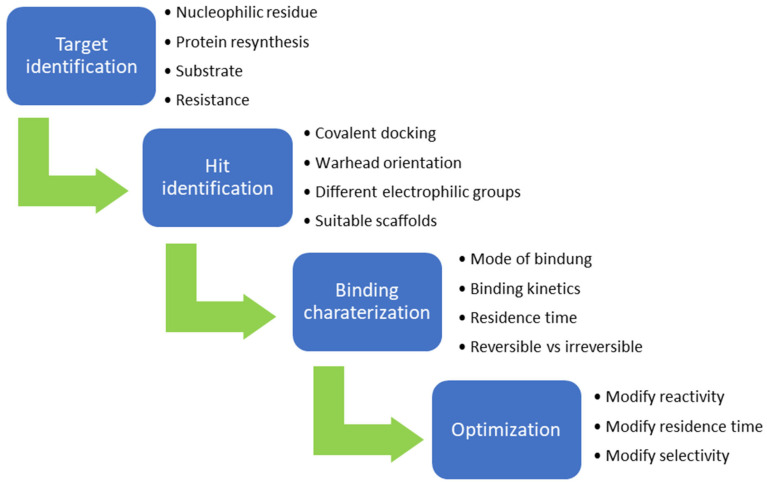
A brief illustration of the step-by-step process for the design of covalent inhibitors.

**Figure 12 pharmaceuticals-16-00663-f012:**
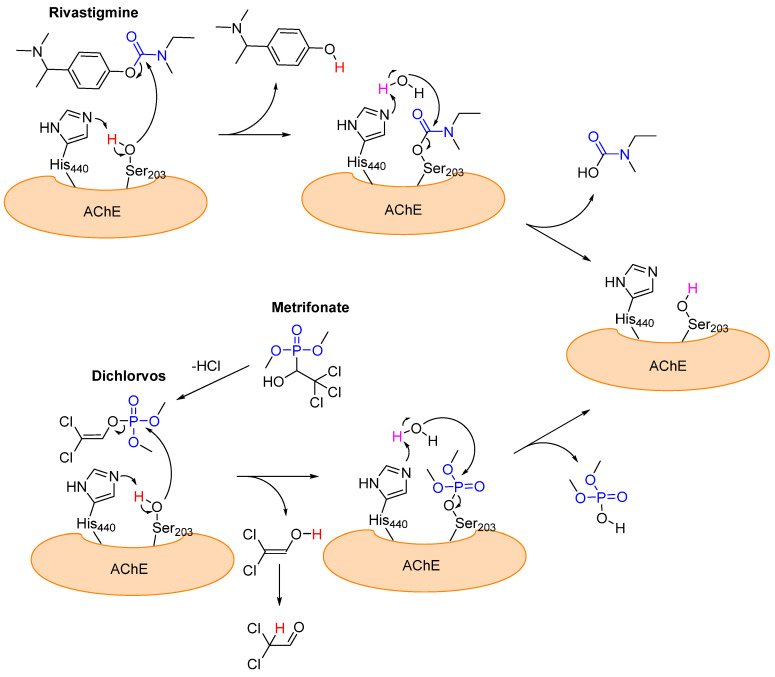
A structural representation of the two covalently binding AchE inhibitors rivastigmine and metrifonate (dichlorvos) and their mechanisms of action with the Ser_203_ from the catalytic triad of AchE. The reactive functional groups are highlighted in blue.

**Figure 13 pharmaceuticals-16-00663-f013:**
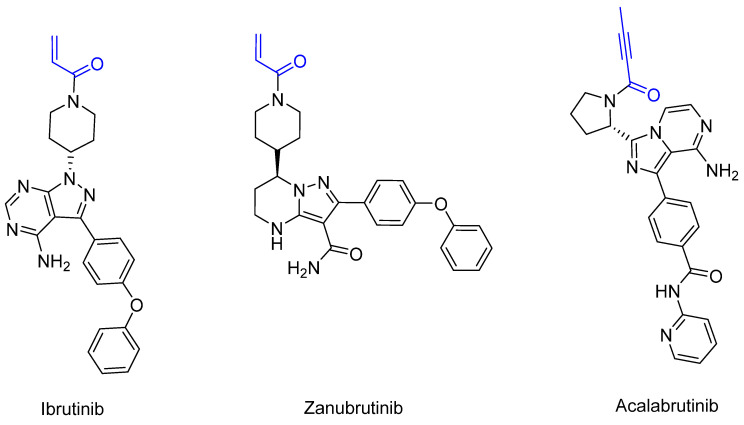
A structural representation of the covalent irreversible BTK inhibitors ibrutinib (**left**), zanubrutinib (**middle**) and acalabrutinib (**right**). The reactive groups (warheads) are marked in blue.

**Figure 14 pharmaceuticals-16-00663-f014:**
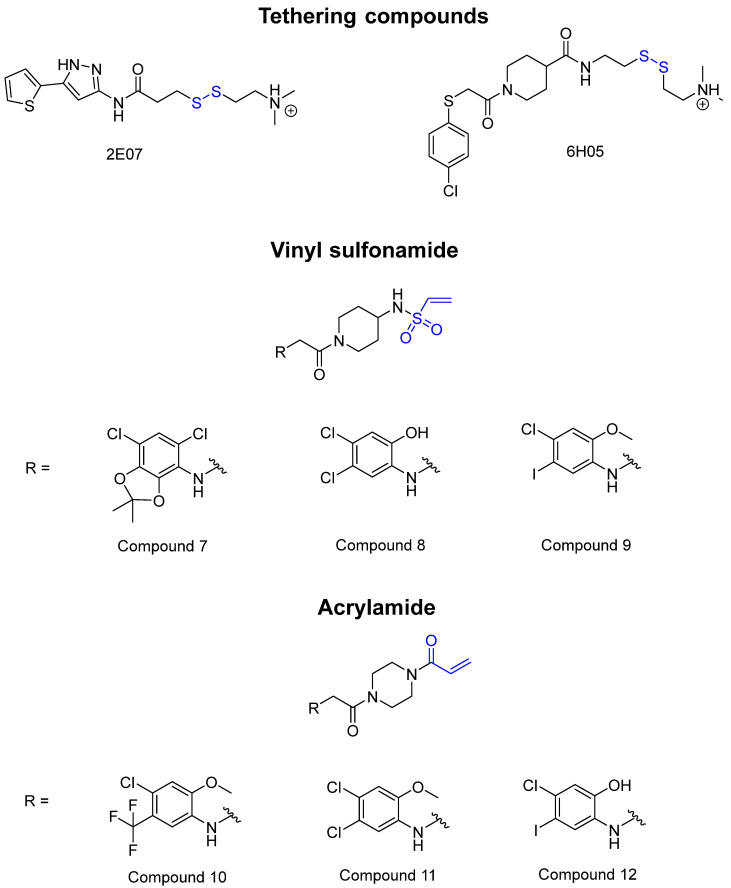
A graphical representation of the reversible tethering compounds 2E07 and 6H05 and an exemplary representation of certain irreversible covalent inhibitors with vinyl sulfonamide or acrylamide structures, produced by Ostrem et al. [85]. The warheads are highlighted in blue.

**Figure 15 pharmaceuticals-16-00663-f015:**
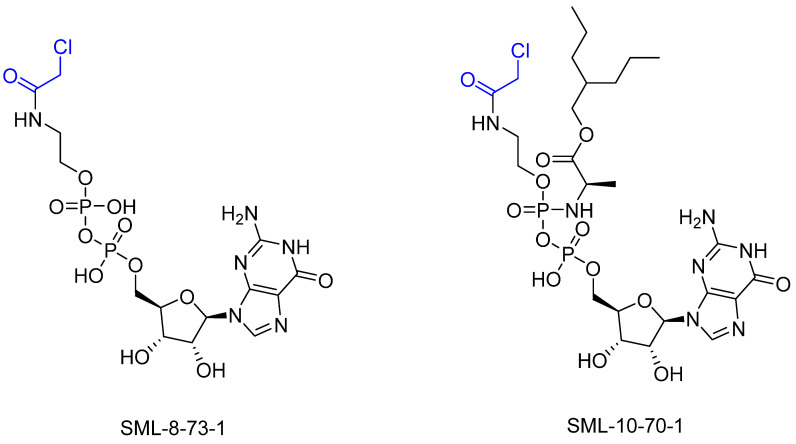
GDP-based inhibitors for G12C SML-8-83-1 (**left**) and SML-10-70-1 (**right**). The warheads are highlighted in blue.

**Figure 16 pharmaceuticals-16-00663-f016:**
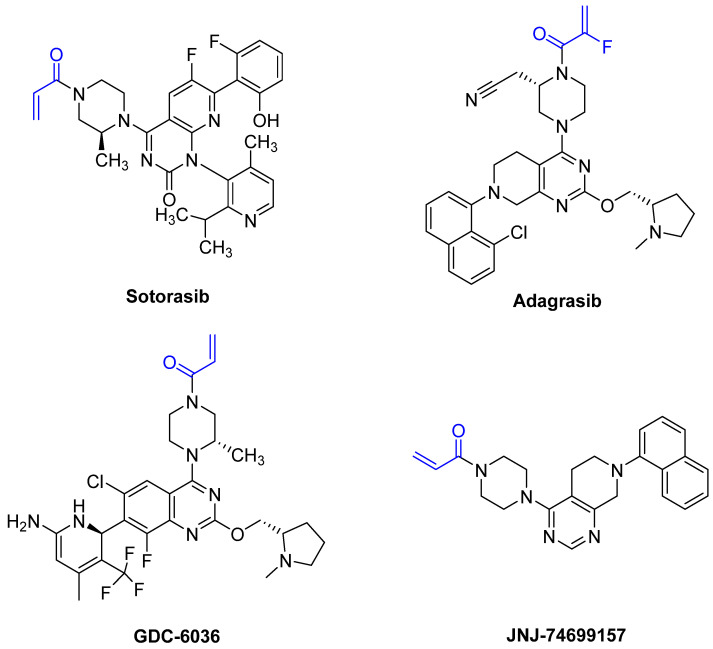
The structure of new G12C inhibitors. The warheads are marked in blue.

**Figure 17 pharmaceuticals-16-00663-f017:**
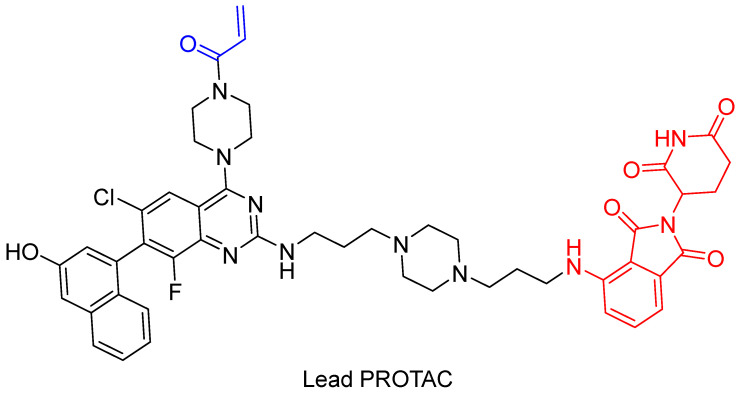
A structural representation of the lead PROTAC from Zeng et al. The warhead for binding G12C is marked in blue, and the pomalidomide for binding cereblon is marked in red.

**Figure 18 pharmaceuticals-16-00663-f018:**
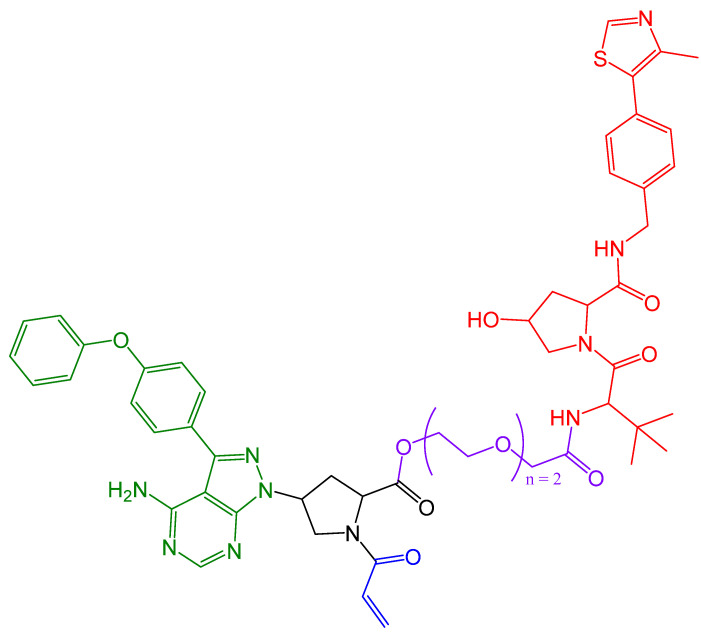
The structure of the BTK PROTAC produced by Xue et al. [114], which was based on an ibrutinib analog. The warhead is marked in blue, the VHL is marked in red, the linker is marked in purple, and the ibrutinib is marked in green.

**Figure 19 pharmaceuticals-16-00663-f019:**
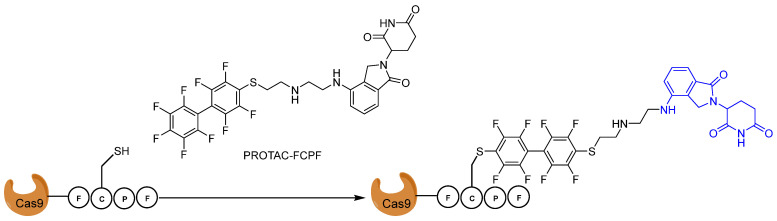
The structural assembly of Cas9^FCPF^ with the PROTAC-FCPF, which was based on lenalidomide (highlighted in blue).

**Figure 20 pharmaceuticals-16-00663-f020:**
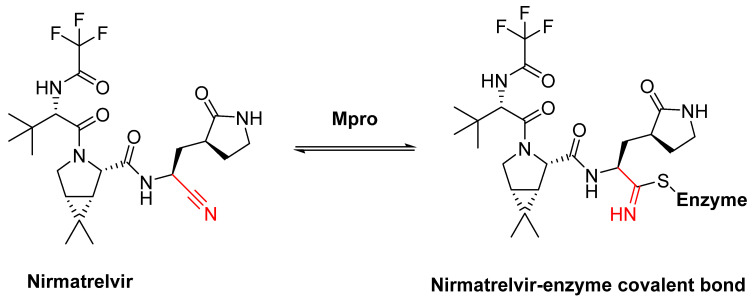
The structure of nirmatrelvir (**left**) and a nirmatrelvir-enzyme covalent bond (**right**). The warhead is marked in red.

**Figure 21 pharmaceuticals-16-00663-f021:**
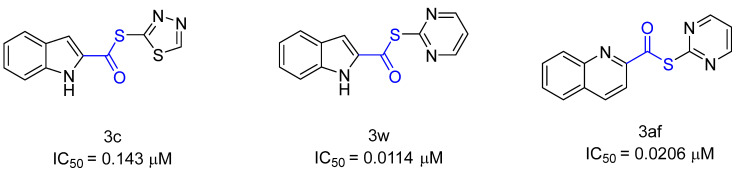
The structure of the covalent M^Pro^ inhibitors by Pillaiyar et al. [123]. The reactive group is highlighted in blue.

**Figure 22 pharmaceuticals-16-00663-f022:**
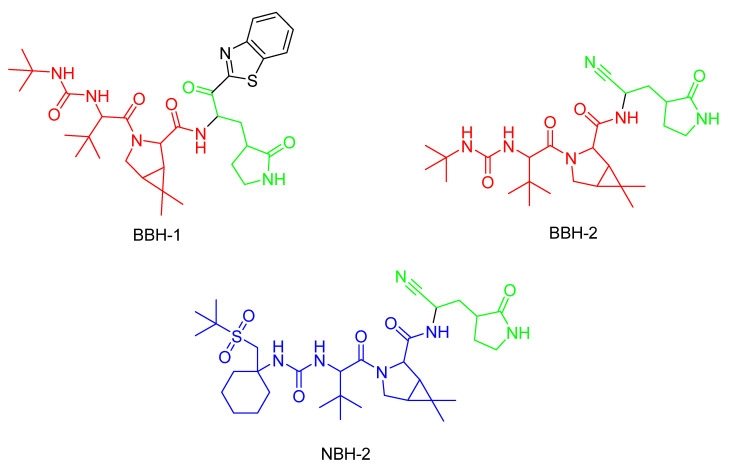
The structure of the SARS-CoV-2 hybrid inhibitors BBH-1 (**left**), NBH-2 (**center**), and NBH-2 (**right**) by Kneller et al. The inhibitors consist of boceprevir (red), narlaprevir (blue), GC-376, and nirmatrelvir (green).

**Table 1 pharmaceuticals-16-00663-t001:** The FDA-approved covalent inhibitors since 2010, with a structural representation of the warhead and a description of the inhibitor’s function. The warhead positions which are responsible for the formation of the covalent bond are shown in red.

Year	Name of Drug	Warhead	Function
2010	Ceftaroline (β-lactam)	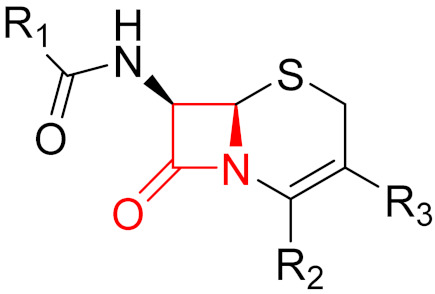	β-lactam antibiotic
2011	Telaprevir (α-ketoamide)	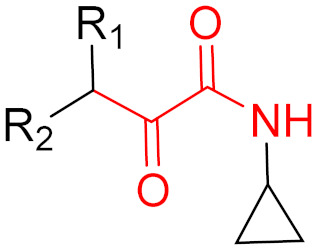	HCV protease inhibitor
2011	Boceprevir (α-ketoamide)	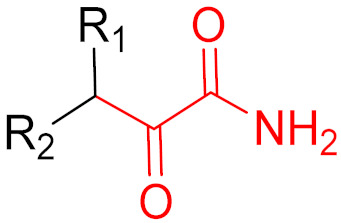	HCV protease inhibitor
2011	Abiraterone (-)	-	Prostate cancer treatment
2012	Carfilzomib (epoxide)	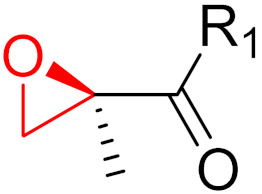	Proteasome inhibitor (cancer)
2013	Afatinib (α,β-unsaturated carbonyl)	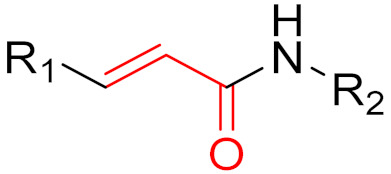	EGFR tyrosine kinase inhibitor
2013	Dimethyl fumarate (α,β-unsaturated carbonyl)	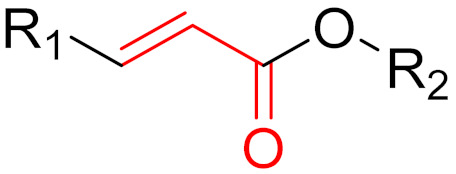	Immunomodulatory drug
2013	Neostigmine (carbonyl group)	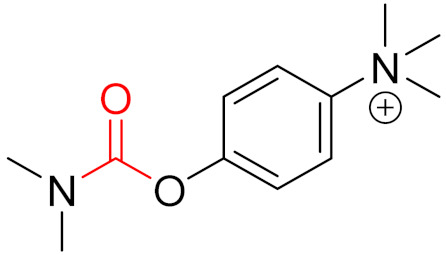	Acetylcholinesterase inhibitor
2013	Ibrutinib (α,β-unsaturated carbonyl)	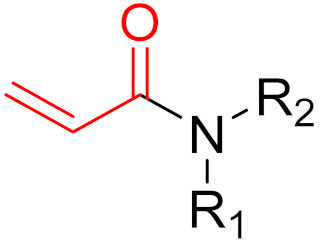	EGFR tyrosine kinase inhibitor
2014	Ceftolozane (β-lactam)	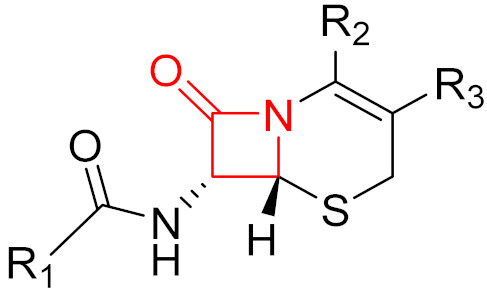	β-lactam antibiotic
2015	Osimertinib (α,β-unsaturated carbonyl)	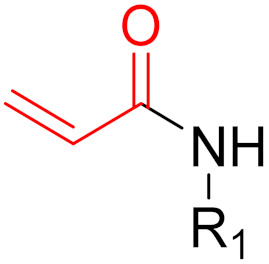	EGFR tyrosine kinase inhibitor
2015	Olmutinib (α,β-unsaturated carbonyl)	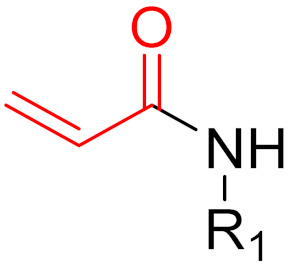	EGFR tyrosine kinase inhibitor
2016	Narlaprevir (α-ketoamide)	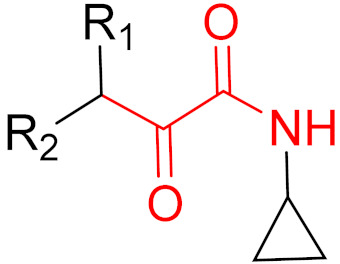	HCV protease inhibitor
2017	Acalabrutinib (α,β-unsaturated propargylamide)	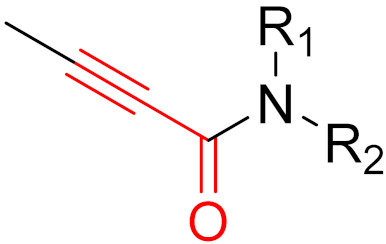	Bruton’s tyrosine kinase inhibitor
2017	Neratinib (α,β-unsaturated carbonyl)	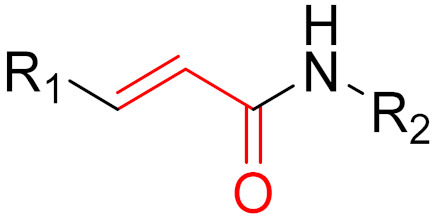	EGFR tyrosine kinase inhibitor
2017	Vaborbactam (boronic acid)	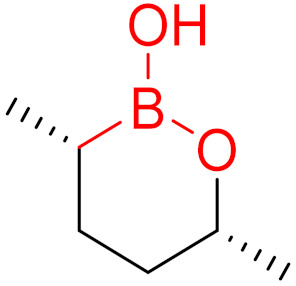	Non-β-lactam β-lactamase inhibitor
2018	Dacomitinib (α,β-unsaturated carbonyl)	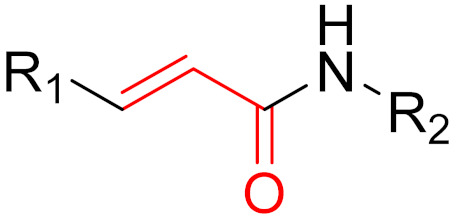	EGFR tyrosine kinase inhibitor
2019	Selinexor (α,β-unsaturated carbonyl)	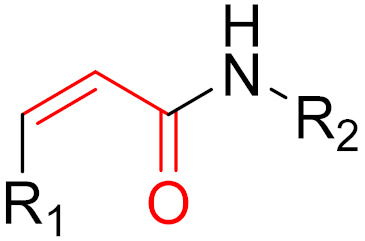	Nuclear export inhibitor
2019	Zanubrutinib (α,β-unsaturated carbonyl)	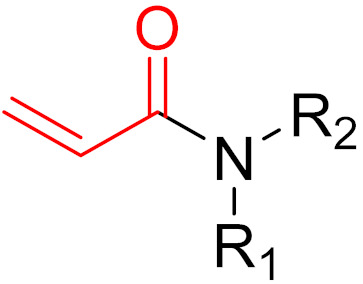	Bruton’s tyrosine kinase inhibitor
2019	Cerfiderocol (β-lactam)	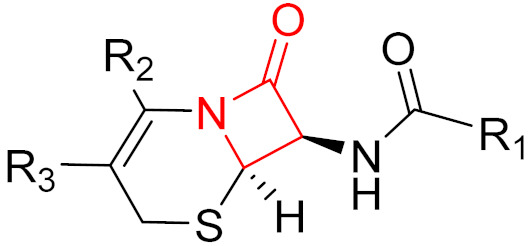	β-lactam antibiotic
2019	Voxelotor (aldehyde)	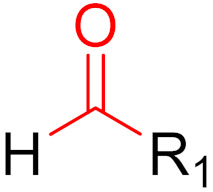	Hemoglobin oxygen-affinity modulator
2021	Sotorasib (α,β-unsaturated carbonyl)	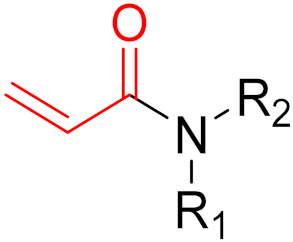	KRAS G12C inhibitor
2021	Nirmatrevir (nitrile)	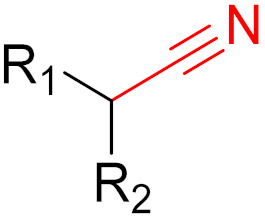	SARS-CoV-2 main protease inhibitor

**Table 2 pharmaceuticals-16-00663-t002:** The advantages and disadvantages of covalent and non-covalent inhibitors.

Type of Inhibitor	Advantages	Disadvantages
**Non-Covalent**	Large non-covalent compound library Easier to evade toxicity in comparison to irreversible covalent inhibitors (long-term inhibition) No need for strong or activated nucleophiles	Comparatively low selectivity Not very potent Limited to non-covalent binding affinity Mostly poor reactivity
**Covalent**	Can be administered at lower doses Higher potency Longer duration of time/inhibition Less sensitive to pharmacokinetic parameters Can provide higher selectivity Higher biochemical efficiency Lower risk of drug resistance Able to target undruggable proteins Binding properties can be influenced by the choice of warhead (i.e., reversible or irreversible)	May cause unexpected toxicity or hypersensitivity May cause drug-induced toxicity The potential immunogenicity of the resulting target adducts The need for strong or activated nucleophiles The need for accessible nucleophile May not be suitable for targets with fast enzyme turnover or fast degradation

## Data Availability

Not applicable.

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
