# Peer review of "Recent Advances in Covalent Drug Discovery"

_pharmaceuticals, 2023, doi:10.3390/ph16050663_

Round 1

Reviewer 1 Report (Previous Reviewer 2)

(1)   A scheme should be provided to illustrate the principle or mechanism for the design of covalent inhibitor.

(2)   A Table should be given to show the types of different covalent reactions for drug designed, and the groups for covalent reactions.

Author Response

Reviewer #1:

Thank you for the helpful feedback.

In this updated version, we conducted a thorough review of the spelling and grammar, taking great care to ensure accuracy. Furthermore, we sought the assistance of MDPI author services to meticulously review the entire manuscript for any errors or inconsistencies.

(1)   A scheme should be provided to illustrate the principle or mechanism for the design of covalent inhibitor.

à We have included a brief illustration of the process in a figure. Thank you for this advice.

(2)   A Table should be given to show the types of different covalent reactions for drug designed, and the groups for covalent reactions.

à Thank you for your professional suggestion! In our manuscript, we have thoroughly reviewed the differences between covalent and non-covalent inhibitors, as well as reversible and irreversible covalent inhibitors. In the revision, we have also included the Michael addition. We acknowledge that this topic is vast and complex, and there are already several detailed publications that exclusively focus on describing these covalent reactions in great depth. Therefore, providing a precise and detailed description within the scope of our publication, considering the available time and space, would not be feasible. We regret that we could not include all the information regarding the various types of covalent reactions in drug discovery and their associated groups. However, we have included two comprehensive publications as additional sources in the text. We aim to meet your expectations and provide a comprehensive overview within the limitations of our manuscript.

Reviewer 2 Report (New Reviewer)

The revision of the ‘Recent Advances in Covalent Drug Discovery’ manuscript has been significantly improved over the original and can be accepted with minor modifications.

Comments and Suggestions for Authors:

Figure 7 - caption.  Please specify, if known, which hepatic cytochrome P450 catalyzes this reaction; oxidation, not oxyidation (!), as stated in Figure 7.

Table 2. Please adjust the indentation and position of the bullet points.

Page 21. 'Compared to molecular glue degraders (typically < 500 g/mol), PROTACs can vary drastically in size, ranging from 500 g/mol to > 1000 g/mol.' I'm guessing it's about molecular weight (MW). Please specify that, e.g. (MW typically < 500 g/mol), ...

Figure 21. In graphics: -S-Enzym, instead of -S-Enzyme (nirmatrelvir-enzyme covalent bond (right)).

Figure 23 - caption. NBH-2(right) - omitted space.

Author Response

Reviewer #2:

The revision of the ‘Recent Advances in Covalent Drug Discovery’ manuscript has been significantly improved over the original and can be accepted with minor modifications.

à We greatly appreciate your generous feedback. We are delighted to hear that our efforts have resulted in an improved manuscript.

Figure 7 - caption.  Please specify, if known, which hepatic cytochrome P450 catalyzes this reaction; oxidation, not oxyidation (!), as stated in Figure 7.

 à Thank you for the important advice! We have specified the types of CYP450 in the metabolism of clopidogrel.

Table 2. Please adjust the indentation and position of the bullet points.

à We modified them according to your suggestion.

Page 21. 'Compared to molecular glue degraders (typically < 500 g/mol), PROTACs can vary drastically in size, ranging from 500 g/mol to > 1000 g/mol.' I'm guessing it's about molecular weight (MW). Please specify that, e.g. (MW typically < 500 g/mol), ...

à We have clarified them in the revision.

Figure 21. In graphics: -S-Enzym, instead of -S-Enzyme (nirmatrelvir-enzyme covalent bond (right)).

 à Thank you for the important advice! We have corrected in the revision.

Figure 23 - caption. NBH-2(right) - omitted space.

à We changed it. Thanks!

Reviewer 3 Report (New Reviewer)

The manuscript entitled “Recent Advances in Covalent Drug Discovery” presented a comprehensive review of drugs containing electrophile moiety irreversibly binding to disease-relevant protein and their mode of action. It is recommended to publish this work if the following comment can be responded, and suggestions can be addressed in revised manuscript.

11)      In figure 2, Warfarin and Tranylcypromine is unlikely a covalent drug or make effects through covalent binding. Please remove/change them unless the author can provide more evidence/references to support that.

22)      ABPP can also be started with cells rather than only lysate. See ref https://www.nature.com/articles/s41587-020-00778-3.

33)      In the part of FBDD, it is recommended to discuss about at least one case of drug discovery from fragment to colvalent molecule such like KRAS inhibitor Sotorasib (AMG 510).

44)      In the part of PROTAC, the author should mention more cases using covalent E3 ligands such as: https://www.nature.com/articles/s41589-019-0279-5; https://pubs.acs.org/doi/full/10.1021/jacs.2c08964; https://pubs.acs.org/doi/full/10.1021/acschembio.8b01083.

55)      The discussion about CRBN ligands and VHL ligands is irrelevant to the covalent drug, the topic of this review. It is recommended to remove or reduce the content of that part as well as figure 16 and 17.

Author Response

Reviewer #3:

The manuscript entitled “Recent Advances in Covalent Drug Discovery” presented a comprehensive review of drugs containing electrophile moiety irreversibly binding to disease-relevant protein and their mode of action. It is recommended to publish this work if the following comment can be responded, and suggestions can be addressed in revised manuscript.

 à Thank you for your work. We hope we could successfully adapt the manuscript!

11)      In figure 2, Warfarin and Tranylcypromine is unlikely a covalent drug or make effects through covalent binding. Please remove/change them unless the author can provide more evidence/references to support that.

à Thank you for the important advice! We have removed the two active ingredients mentioned and replaced them with covalent active ingredients.

22)      ABPP can also be started with cells rather than only lysate. See ref https://www.nature.com/articles/s41587-020-00778-3.

à We changed it.

33)      In the part of FBDD, it is recommended to discuss about at least one case of drug discovery from fragment to colvalent molecule such like KRAS inhibitor Sotorasib (AMG 510).

à Thank you for the important advice! We mentioned sotorasib in the revision.

44)      In the part of PROTAC, the author should mention more cases using covalent E3 ligands such as: https://www.nature.com/articles/s41589-019-0279-5; https://pubs.acs.org/doi/full/10.1021/jacs.2c08964; https://pubs.acs.org/doi/full/10.1021/acschembio.8b01083.

à  We appreciate your comments and suggestions. We have summarized E3 ligases used for the synthesis of PROTACs in the revision.

55)      The discussion about CRBN ligands and VHL ligands is irrelevant to the covalent drug, the topic of this review. It is recommended to remove or reduce the content of that part as well as figure 16 and 17.

à We understand your comment and have removed the respective figures as well as a large part of the comparison between VHL and CRBN.

This manuscript is a resubmission of an earlier submission. The following is a list of the peer review reports and author responses from that submission.

Round 1

Reviewer 1 Report

This reviewe provideds a comprehensive summary on the history and current progress on the development of covalent drug discovery, which is an emerging topic in medicinal chemist community. The manuscript was generally well prepared. However, there are some issues need be solevd before it is publishable. 

1) Figue 1. the warheads foe covalent (irrevesible or reversible) are not completed. Please update the useful wards for the covalent inhibitor design.

2) line 76. "DPP-4 in turn degrades the hormone glucagon-like pep- 76 tide 1 (GLP-1), promoting the release of insulin." is misleading. Degradation of GLP-1 does not promot the release of insulin.

3) the statement of that " covalent-binding agents are usually less suscep- 104 tible to drug resistance caused by mutations in chemotherapy. 6, 25 As long as the binding 105 occurs, inhibition is effective and the risk of resistance is relatively low" is incorrecct. For instance C797S mutation is one of major mechanisms contributing the resistance against the 3rd generation EGFR inhibitor therapy.

4) the authors discussed about the progress on k-Ras G12C inhibitors. However, the literatures they utilized are too old! in fact, an inhibitor has been approved and mutiple wre in clinical trials. The authors still talked about the molecules which were published about 10 years ago. same things happened for the COVID-19 drugs.

5)the authors discussed about the advatage about covalent PROTACs. However, the major disadvatage about irreversible PROTACs is that equivalent amount of PROTACTs will be consumed for molecules with this kind of MOA. While one character of PROTACs is that they may achieve efficacy in catalytical amount. 

6) the authours are suggested to provide a table for comparison of advatage or disadvatage between nocovalent drugs and covalent drugs. 

Author Response

please see attached response to review

Reviewer 2 Report

This work reported the recent advantages of covalent inhibitor. It is interesting for the drug researcher. It could be accepted after the following revisions.

(1)   How to regulate the targeting of covalent inhibitor? This should be summarized and discussed.

(2)   The principle for the design of covalent inhibitor should be discussed.

(3)   The types of different covalent reactions for drug designed, and the groups for covalent reactions should be reviewed.

Author Response

please see attached response to review

Reviewer 3 Report

The review contributed by Chen’s group described the recent advantages in covalent drug discovery. In general, the manuscript was well-written. In this review, a short history of the development of covalent drug was described, also some selected examples were described to explain this strategy. Despite this, the logic for the selection of the diseases and examples from numerous publications showed in Figure 3 in the manuscript was not clear. Moreover, the inflection points of the number of publications appeared around 2012-2014, which was earlier than the COVID-19. What’s the reason behind this? There are also some respects need to be improved:

1)     Some of the fonts are too small to read in Figure 1. and 2.

2)     In Table 1, it is better to draw the structures and the reactive sites of the drugs.

3)     In line 116, the “Figure 5” should be replaced with “Figure 6”.

4)     The size of the structural formula was not identical in all the figures.

5)  It seems all the names of journal were missed in the references.

Author Response

!

Round 2

Reviewer 3 Report

The paper has been revised and is suitable for publication. However, some minor problems need to be corrected:

The blue markers in Figure 6 does not mark the correct warhead;

The symbol ')' is missing from line 25;

The first letter of "Resulting" in line 182 needs to be lowercase.

Author Response

Thanks to all reviewers for the critical comments and suggestions that helped us improve the manuscript!